# Application of Polyphenol-Loaded Nanoparticles in Food Industry

**DOI:** 10.3390/nano9111629

**Published:** 2019-11-16

**Authors:** Danijel D. Milinčić, Dušanka A. Popović, Steva M. Lević, Aleksandar Ž. Kostić, Živoslav Lj. Tešić, Viktor A. Nedović, Mirjana B. Pešić

**Affiliations:** 1Department of Food Technology and Biochemistry, Faculty of Agriculture, University of Belgrade, Nemanjina 6, 11080 Belgrade, Serbia; danijel.milincic@agrif.bg.ac.rs (D.D.M.); dusanka.popovic@agrif.bg.ac.rs (D.A.P.); slevic@agrif.bg.ac.rs (S.M.L.); akostic@agrif.bg.ac.rs (A.Ž.K.); vnedovic@agrif.bg.ac.rs (V.A.N.); 2Faculty of Chemistry, University of Belgrade, Studentski Trg, 12-16, 11158 Belgrade, Serbia; ztesic@chem.bg.ac.rs

**Keywords:** nanotechnology, nanoparticles, nanomaterials, polyphenols, food processing, food packaging, nanofood, functional food

## Abstract

Nanotechnology is an emerging field of science, and nanotechnological concepts have been intensively studied for potential applications in the food industry. Nanoparticles (with dimensions ranging from one to several hundred nanometers) have specific characteristics and better functionality, thanks to their size and other physicochemical properties. Polyphenols are recognized as active compounds that have several putative beneficial properties, including antioxidant, antimicrobial, and anticancer activity. However, the use of polyphenols as functional food ingredients faces numerous challenges, such as their poor stability, solubility, and bioavailability. These difficulties could be solved relatively easily by the application of encapsulation. The objective of this review is to present the most recent accomplishments in the usage of polyphenol-loaded nanoparticles in food science. Nanoparticles loaded with polyphenols and their applications as active ingredients for improving physicochemical and functional properties of food, or as components of active packaging materials, were critically reviewed. Potential adverse effects of polyphenol-loaded nanomaterials are also discussed.

## 1. Introduction

Nanotechnology found a wide range of applications in the food sector [1,2], which includes food safety and quality, the targeted delivery of important compounds and increased bioavailability, the development of new products, the control of technological processes, packaging, and sensory improvements, such as texture and taste modifications [3,4,5] (Figure 1).

Nanodrugs and nanopharmaceuticals were developed first, but the applications of nanotechnology in food production and agriculture are rapidly increasing [6]; both food industry and consumers, as well as developing and developed countries, may benefit from them [7]. One of the main roles of nanotechnology in the food sector is the effective transport of important substances that have nutritional and generally beneficial effects to humans [8,9].

A novel approach in nanotechnology is green nanotechnology, which encourages environmental sustainability and green routes of synthesizing nanomaterials and minimizes costs and potential environmental risks. This approach emphasizes the usage of ecofriendly and biocompatible processes, usually with plant extracts or micro-organisms as active compounds or carriers [10,11]. Green nanotechnology is aimed toward the solution of modern problems and topics such as renewable energy, waste management, environmental remediation, and replacing hazardous processes and constituents [12]. The food sector could have huge benefits from adopting green nanotechnology, especially in those processes that are less energy-efficient or actively damage food compounds.

**Figure 1 nanomaterials-09-01629-f001:**
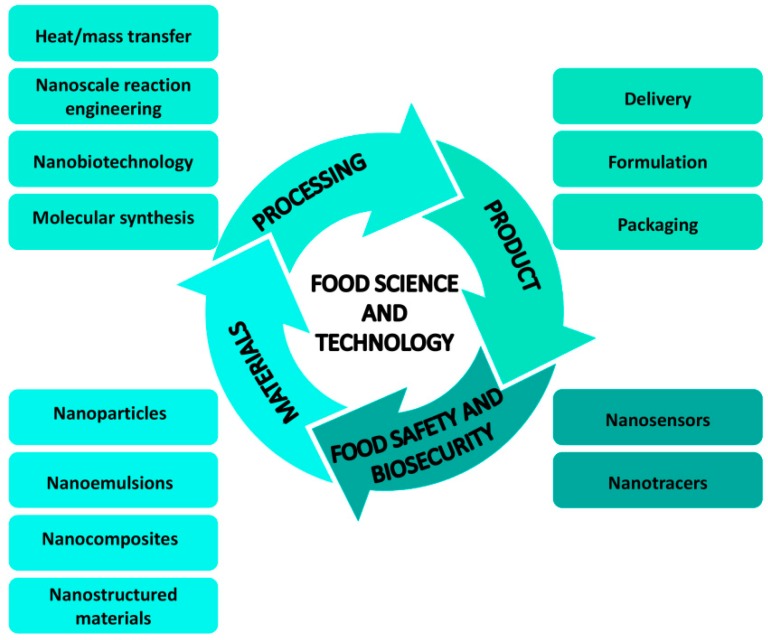
Schematic view of use of nanomaterials in food science and technology (adapted from Ramachandraiah et al. [13], with permission from publisher Asian-Australasian Journal of Animal Science, 2015).

The latest nanotechnology applications in food science include the development of functional foods and nanosized food compounds, the formation of delivery systems for bioactive compounds, and innovations in food packaging [14,15,16,17]. Both microencapsulation and nanoencapsulation aim to improve the functional properties of bioactive compounds or to effectively transport them to the target destinations [2].

There are numerous biologically active compounds that can be encapsulated, but for the food sector, the encapsulation of polyphenols is particularly important [2].

Polyphenols are widely studied secondary plant metabolites due to their potential positive effects on human metabolic processes [18,19,20,21,22,23,24,25,26,27,28,29]. Phenolic compounds have significant antioxidant and antimicrobial characteristics, but they are very unstable and susceptible to degradation processes, poorly soluble, and, in most cases, their bioavailability is relatively low [2]. So, incorporation of plants’ phenolic compounds in food can significantly affect their physicochemical properties, stability, solubility, and bioavailability [1,30]. Encapsulation can prevent the degradation of phenolic compounds due to its protective effects against the negative impact of light, oxygen, and pH. Encapsulation can also prevent interaction between polyphenols and other food components [31]. Further, polyphenol-loaded nanoparticles can be used for the development of new types of functional food, of which the consumption can help prevent various diseases [32].

Different nanocarriers for phenolic components can be used as a protective barrier, and they can roughly be divided into polysaccharide- and protein-based delivery systems [33]. Substances such as cyclodextrins, polymeric nanoparticles, nanomicelles [2], gelatin nanoparticles and films [34,35,36], food-protein nanoparticles like casein, whey proteins, soybean proteins [37,38], zein nanoparticles [39], chitosan [34,40], lipid nanocarriers [41,42], or protein–polysaccharide complex nanoparticles [43] have proven to be suitable carriers for the nanoencapsulation of polyphenolic compounds. The choice of carrier material affects the physicochemical characteristics of the encapsulated substances [44]. Food-protein nanoparticles and chitosan are the most commonly used nanocarriers for the delivery of plant polyphenols because there is solid evidence that they improve the intestinal absorption of phenolic compounds [34]. However, use of chitosan may be limited in the food industry because it has low solubility at neutral pH conditions and low delivery efficiency for individual components [45].

Recently, polysaccharide–protein nanocarriers have been reported as being promising for polyphenols encapsulation [46,47,48]. Active components bind to the protein part of the nanocarrier via hydrogen bonding and hydrophobic interactions, while polysaccharides contribute to the prevention of enzymatic protein degradation in gastric conditions [49]. Polysaccharide–bioactive peptide nanoparticles can also be valuable nanocarriers for the encapsulation of small molecular polyphenols, providing better bioavailability of these valuable components [50].

Among polyphenols, the most commonly encapsulated are catechins, quercetin, eugenol, epigallocatechin, epigallocatechin-gallate (EGCG), curcumin, and polyphenols derived from teas or essential oils (Table 1).

Polyphenols can interact with nanocarriers. For example, it has been shown that there is a certain level of hydrogen bonding between catechin and unreacted amino groups of chitosan in their encapsulates [56]. Catechin can also interact with proline-rich proteins [2,77] or with the phenolic ring of EGCG with prolines in β-casein [2,78]. The mechanisms of encapsulation enabling polyphenol–nanocarrier interactions include ionic gelation, coacervation, liposome entrapment, inclusion complexation, co-crystallization, nanoencapsulation, freeze-drying, yeast encapsulation, and emulsion [79]. These various mechanisms can increase the bioavailability of polyphenols, prevent degradation in the gastrointestinal tract, enhance the delivery of polyphenols directly to the targeted sites [33], or provide stability during storage or processing [80].

The use of encapsulation technologies for the preparation and application of polyphenol-loaded microparticles in the food industry has recently been reviewed [81], but a similar review on polyphenol-loaded nanoparticles has not yet been published. Considering the significance of polyphenols and benefits from their nanoencapsulation, the aim of this paper is to give an overview of recent achievements in the application of polyphenol-loaded nanoparticles in food processing and packaging. The possible toxicity of nanomaterials is also discussed.

## 2. Use of Polyphenol-Loaded Nanoparticles in Food Processing

Polyphenol-rich nanoparticles can be used in food processing for the improvement of physicochemical characteristics, such as food stability, texture, and flavor profile, or functional properties such as antioxidant or antimicrobial activity (Table 1). Nanoencapsulation has become an important approach for preserving the properties of active components during food processing and consumption [82]. Further, nanotechnology-based sensor systems have been developed as alternatives for conventional analytical methods in safety and quality control [83].

### 2.1. Polyphenol-Loaded Nanoparticles for the Enhancement of Physicochemical Properties of Food

Encapsulation is a widely used technique for preserving different food properties [84]. The encapsulation of various substances provides solutions for numerous problems in the food sector, such as the improvement of organoleptic properties, by masking undesirable flavors, odors, and colors; control of volatility, solubility, stickiness, release of active compounds [85], and degradation; and the reduction of possible interactions with other food components, or with light and oxygen [31]. The first physical or chemical property to which consumers pay attention is color, and, because of its impact, it is an important characteristic for both science and industry.

Since polyphenols are a diverse group of plant metabolites, the encapsulation systems used for their protection might be different, too. Besides variations in chemical properties, the criteria for establishing appropriate encapsulation procedures also depend on required encapsulation properties (e.g., particle size, solubility, and controlled release).

Curcumin is a polyphenol present at the rhizomes of the *Curcuma longa* plant; generally, it serves as a distinct yellow–orange natural coloring for food [31] and has a strong impact on food flavor, but it is poorly soluble in water [31,52,86]. Its weak bioavailability and instability under neutral and alkaline pH conditions [31,87] limit its use in food. These problems can be solved by nanoencapsulation, and its potential use in food nanotechnology may bring novel functional foods with improved technological and sensorial properties. Nanoencapsulation could provide products with antioxidant and antimicrobial ingredients, or nanosized coloring agents and substances with improved water solubility [88]. Ranjan et al. [5] and Silva et al. [51] showed that, besides the use of nanomicelle-encapsulated curcumin for color improvement, these nanomicelles, at the same time, allow for its better intestinal bioavailability and enhanced stability (Table 1).

The use of native casein micelles (CM) as nanocarriers of biologically active compounds has recently gained attention. CM structure depends on the type of milk and can easily be modified by processing parameters [89,90,91,92], making CM an ideal platform for the delivery of bioactive compounds. It has been reported that β-carotene [93] and vitamin D [94] can be successfully loaded into native CM, enabling their improved storage stability and bioavailability. It was demonstrated that the addition of tea polyphenols to milk affects the technological properties of casein micelles: decreased gelling properties [95] and increased heat stability [96]. Knowing that milk is generally recognized as a source of beneficial substances for the growth and health of children and adults with the verified GRAS status of casein [97], it is worth trying to optimize the polyphenol nanoencapsulation process. Phenolic plant compounds could be extracted from byproducts of the food industry and further encapsulated in milk-based carriers, or other suitable protective materials using a spray-drying process. Spray drying is a commonly used method for the production of large quantities of encapsulates for the food industry. Besides the usage of elevated temperatures, it was shown that spray-drying preserves the main antioxidant properties of plant phenolic compounds [44]. Combining conventional encapsulation methods, such as spray drying, with newly developed nanocarriers could lead to the development of a new generation of dairy products, providing additional benefits to human health.

### 2.2. Polyphenol-Loaded Nanoparticles for the Enhancement of Functional Properties of Food

#### 2.2.1. Antioxidant Properties

The aim of polyphenolic-nutrient nanoencapsulation is to protect and transport these active components and ensure their maximum bioavailability. For example, the dominant polyphenols of tea catechin and its derivatives (−)-epigallocatechin-3-gallate, (−)-epicatechin-3-gallate, (−)-epigallocatechin, and (−)-epicatechin, have strong biological properties, such as antioxidant and anticancer activities [36,45,98]; on the other hand, if they are not encapsulated, their bioavailability is low [98]. Consequently, there is a necessity for the development of effective delivery systems for improving of polyphenolic components’ bioavailability.

In the study of Tang et al. [54], chitosan/γ-PGA (PGA-edible polypeptide (poly-γ-glutamic acid)) nanoparticles were prepared for the oral delivery of tea catechins. Briefly, tea-catechin-loaded nanoparticles showed very effective radical scavenging capacity and could serve as suitable carriers for the transmucosal delivery of tea catechins. According to the same authors, nanoencapsulated catechins could be used as food additives and dietary supplements.

It has been found that, among tea polyphenols, EGCG has the largest scavenging capacity [99,100], but in many studies, it has been indicated that EGCG has low bioavailability because of its large-sized molecules and the number of hydrogen bonds [100,101]. Application of chitosan/β-lactoglobulin nanocarriers provided sustained release of tea EGCG and thereby increased its absorption in the human intestine [45]. Nanocarriers such as dextran sulfate, combined with N,N-dimethylhexadecyl carboxymethyl chitosan nanoliposome (DS-DCMC-NL), are also a suitable carrier for EGCG, which possesses a high encapsulation capacity and the potential for further application in food processing [102].

In addition to tea-polyphenol-loaded nanoparticles, there are studies that examined the nanoencapsulation of other polyphenol nutrients that exert antioxidant activity, such as curcumin [5,43,51,52], quercetin [58,59,60], rutin [49], and naringenin [2,62]. According to Chang et al. [52], the use of pectin-coated NaCas/zein nanocarriers for curcumin encapsulation significantly improved its antioxidant activity and prolonged its release capabilities in simulated gastric and intestinal fluids, whereas quercetin-loaded solid-lipid nanoparticles improved its bioavailability [60]. The determination of the antioxidant properties of polyphenol-loaded nanoparticles was done using different antioxidant assays, such as ABTS/DPPH radical scavenging activity, hypochlorous acid/hydrogen peroxide scavenging assays, and ferric-reducing ability (Table 2).

Hydroxycinnamic acids (HAs; ferulic, caffeic, synaptic, and coumaric acid) were successfully incorporated into lipid-core nanocapsules that can be used for the production of functional foods. This delivery system was able to preserve HAs in simulated gastric fluids and to enable their release in the simulated intestinal fluid [113]. Simulated in vitro gastrointestinal digestions are a valuable tool for determining the bioaccessibility of polyphenols, and scientists researching the topic of polyphenol-based food nanocapsules must take into consideration the effect of polyphenol co-digestion with different food components and the complex interactions between polyphenols and a food matrix [30]. In the study of Pešić et al. [30], enrichment of meat- and cereal-based products with grape polyphenol extracts appeared to be beneficial, but a review by Silva et al. [114] outlined both advantages and limitations of nanoencapsulated polyphenols when added to dairy beverages, since interactions with the food matrix may alter the bioavailability of polyphenols and even devitalize their function.

#### 2.2.2. Antimicrobial Properties

Frequent foodborne-disease outbreaks are one of the main concerns of the food industry. This is one of the reasons why consumer demands for the use of natural components have increased, which leads to the improvement of microbiological safety. Incorporation of components that possess antimicrobial activity into adequate nanocarriers can help control or prevent the growth of pathogens and spoilage micro-organisms [115]. It is assumed that polyphenols exhibit an antimicrobial effect because of their absorption to cell membranes and interaction with enzymes [32]. Eugenol belongs to the group of phenolic components that possess strong antimicrobial and antioxidant capacity [116], but it has low water solubility [48,69], and its applications are limited. Stable eugenol-loaded zein/caseinate/pectin complex nanoparticles obtained through a heat- and pH-induced complexation process could find application in the food industry as dry-powder formulation with antimicrobial properties [48]. In a study by Ghosh et al. [68], sesame-oil-blended eugenol-loaded nanoemulsion formation, as well as properties of obtained nanoemulsion, such as droplet size and stability, depended on the type of surfactant. This emphasizes the need for proper emulsion formation/stabilization using adequate emulsifiers. Nanoemulsion formed in this way exhibited pronounced activity against *S. aureus* and affected the reduction of heterotrophic bacteria in orange juice.

It has been shown that thymol acts on Gram-positive and -negative bacteria [65,117] and has a strong impact on food flavor, but it is poorly soluble in water [65]. Thymol-loaded zein nanoparticles stabilized with sodium caseinate and chitosan hydrochloride obtained by liquid–liquid dispersion showed strong antimicrobial activity against *S. aureus* and could potentially be used as delivery systems for antimicrobial agents in food products [65]. In an earlier study performed by Hu et al. [98], thymol-loaded chitosan nanoparticles showed stronger activity against Gram-positive bacteria compared to Gram-negative bacteria.

Formation of a nanocomplex between chitosan and different phenolics, such as rosmaric acid, protocatechuic acid, and 2,5-dihydroxybenzoic acid, against food pathogens has been described by Madureira et al. [32]. Briefly, polyphenol-loaded chitosan nanoparticles had better antimicrobial activity against *Escherichia coli O157:H7* and *Bacillus cereus,* while, against *Salmonella Typhimurium,* they had lesser activity. The obtained images by electronic microscopy showed mechanisms of bacterial-cell damage. After polyphenol adsorption, holes on the surface of the cell were noticed, and even lysis was possible. Other antimicrobial components in the nanoencapsulated forms could be even more efficient against pathogens. For example, natural antimicrobials from propolis showed better antimicrobial activity when they were in the form of nanosized particles [118].

#### 2.2.3. Health-Promoting Properties

Besides antimicrobial properties, the potential health benefits of polyphenols were in a much broader scope of numerous studies in recent years. The beneficial effects of resveratrol, such as antioxidant, antitumor, anti-inflammatory, and antiaging effects, can be improved by nanoencapsulation [119]. However, among numerous investigations, only few in vivo studies were performed to confirm the health-promoting properties of nanoencapsulated resveratrol, indicating that further research is required [120]. Curcumin also showed antidiabetic properties when curcumin–metal (ZnO) complexes were loaded with chitosan [112]. The addition of curcumin nanocapsules in food also resulted in improved functional properties (better solubility and stability, and lower degradation rate) while keeping its remarkable biological functions [121]. The study of Di Costanzo and Angelico [122] reviewed the numerous possibilities of silymarin nanoencapsulation, a mixture of flavonolignan and flavonoid polyphenolic compounds from the *Silybum marianum* seed, which has significant biological potential (antioxidant, anti-inflammatory, anticancer, and antiviral activities).

The synergistic effect of polyphenols is being studied, and it is known that certain polyphenol combinations have better health-promoting characteristics when combined rather than with the sum of individual ones [123].

Nanogels where tea polyphenols were encapsulated with lysozyme and carboxymethyl cellulose showed antitumor effects and represent a promising carrier for functional food ingredients [103]. There are numerous studies pointing out the useful biological functions of green-tea polyphenols: anti-inflammatory, antiproliferative, antihypertensive, antithrombogenic, and lipid-lowering effects [100], emphasizing the role of epicatechin [124], catechin [125], and other polyphenols that may find their function in food nanotechnology. However, the majority of conducted investigations were to estimate the anticancer activity of polyphenol-loaded nanoparticles on different cancer cells, such as hepatocyte, kidney, colon, oral, breast, and cervix cells (Table 2).

Further, it should be emphasized that the beneficial effects of polyphenols were claimed in the literature mostly on the basis of in vitro studies, but were not demonstrated in in vivo studies. For example, a relationship has not been established between catechin consumption from green tea (*Camellia sinensis* (L.) Kuntze) and the maintenance of normal blood cholesterol concentrations, normal blood pressure, normal vision, normal bones, or the protection of DNA, proteins, or lipids from oxidative damage [126]. The same conclusion was reported for olive-oil polyphenols and the maintenance of normal blood HDL cholesterol concentrations [127], or for flavonoids from fruit juices and antioxidativity, and flavonoids from citrus and vascular health [128].

## 3. Use of Polyphenol-Loaded Nanoparticles in Food Packaging

Packaging is a very significant operation in the food industry. Nanotechnology plays an important role in research and innovation in packaging technology, such as smart and active packaging [129,130,131]. Nanoparticle incorporation into packaging materials improves their barriers (gas/water/aroma), mechanical or oxygen-scavenging properties, thus enhancing food quality and prolonging the shelf-life [82]. In addition to improving the physical properties of packaging materials, nanoparticles add other functions, such as antimicrobial and antioxidant properties, or act as nanosensors/nanobiotracers, enabling better food security [82,132].

There is a growing demand for the development of new environmentally friendly films for food packaging based on biomaterials [133]. Nanofilms compared to traditional food-packaging films should have another dimension: the slow and controlled release of antimicrobial and other active components [134]. This is especially important in the case of antioxidants that, as components of packaging materials may provide prolonged shelf-life for food products (Table 3). Nanoemulsions containing polyphenols have been used for creating fruit-based edible films, which showed not only good antimicrobial and physicomechanical properties (such as permeability to water vapor), but also had an environmentally friendly dimension thanks to waste usage from the food industry. Pectin/papaya puree/cinnamaldehyde nanoemulsion edible composite films could be environmentally friendly antimicrobial packaging material for food applications [133].

It has also been demonstrated that the antibacterial properties of nanoencapsulated cinnamaldehyde used for food coatings were better when compared with free cinnamaldehyde. These films showed significant activity against *E. coli* and *B. cereus* [134].

Besides coating, food shelf-life could be prolonged with protection against the oxidation of fatty foods, using polyphenols such as EGCG in the form of nanosized particles [135]. A similar effect was obtained with green-tea polyphenol-loaded nanoparticles for the same purpose [136]. The incorporation of sage extract (*Salvia officinalis* L.) to poly (ε-caprolactone) films resulted in the formulation of potential food-packaging material with both antioxidative and antimicrobial functions [137].

## 4. Nanomaterial and Polyphenol Toxicity

Contrasted with the benefits of nanotechnology, the potential risks this application could bring should be considered. Although there is still no evidence on the health risks of polyphenol-loaded nanoparticles, the risk assessment of nanomaterials, such as nanosilver, zinc oxide, or silicon dioxide, used for their encapsulation, was conducted and well-reviewed [132]. On the other hand, the potential toxicities of polyphenols were also reported in the literature, along with their health benefits [138].

### 4.1. Nanomaterial Toxicity

The EU Commission Recommendation’s definition of the term “nanomaterial” is based only on the size of the nanoparticle, without dealing with its potentially hazardous characteristics (2011/696/EU). Nanotoxicology deals with the adverse effects resulting from exposure to nanomaterials with hazard potential [139]. The key physicochemical properties of nanoparticles that are responsible for toxicity include particle size, surface area and reactivity, crystal structure, aggregation abilities, composition/surface coatings, synthesis and preparation modifications, and purity of sample [140]. Special concerns refer to the uses of engineered nanoparticles (ENPs) in food-packaging materials, such as nanosilver, silanated silicon dioxide, titanium dioxide, iron oxide, or zinc oxide, and their possible migration into foodstuff. The migration is influenced not only by the physicochemical properties of ENPs but also by environmental conditions, type of food, characteristics of packaging materials, position of the ENPs in the packaging materials and their interactions, and contact time, which were recently well-reviewed by Enescu et al. [132]. These authors also gave an overview of current regulations on active food and beverage packaging, standard methods, and analytical techniques to monitor the overall and specific migration of ENPs, as well as their potential health risks.

**Table 3 nanomaterials-09-01629-t003:** Nanoencapsulation of phenolics important for food packaging.

Active Compounds	Nanocarriers	Particle Size (nm)	Activity (Details of Research)	Reference
Catechin (CAT); Catechin-Zn complex (CAT-Zn)	β-chitosan NPs (β-CS NPs)	208–591 nm	Both CAT and CAT-Zn complex-loaded β-CS NPs exhibited a strong antibacterial activity against *E. coli* and *L. innocua*, and they can be used as food supplements or for incorporation into food-packaging materials.	Zhang et al. [79]
Carvacrol	Starch and gelatinized starch	495–529 nm	Carvacrol increased the flexibility, solubility, water vapor permeability, antioxidant, and antimicrobial activity of formed dispersion films, so they can be used as bioactive films.	Homayouni et al. [141]
Eugenol	SiO_2_-eugenol liposome	315.7 ± 0.7 nm	SiO_2_-eugenol liposomes have stabile and pronounced antioxidant activity during 60 days of storage. SiO_2_-eugenol liposome-loaded electrospun nanofibrous membranes showed strong antioxidant activity on beef, and, in the future, they can be used for food preservation.	Cui et al. [142]
Cinnamaldehyde	Nanoliposomes (lipid bilayers of polydiacetylene-N-hydroxysuccini-mide)	100–400 nm	Nanoencapsulated cinnamaldehyde immobilized on glass surfaces showed significant antimicrobial effect, and, in the future, it can be used as an active packaging material for preserving liquid foods.	Makwana et al. [134]
Cinnamaldehyde	Pectin/papaya puree nanoemulsion	20–500 nm	Edible films for food packaging containing small droplets of polyphenol-loaded nanoemulsion had pronounced antimicrobial effect, because encapsulated cinnamaldehyde showed significant antimicrobial properties against food pathogens such as *E. coli*, *S. enterica*, *L. monocytogenes*, and *S. aureus*.	Otoni et al. [133]
Green tea extracts	Hydroxypropyl-methylcellulose (HPMC) containing polylactic acid (PLA) NPs	47–244.4 nm	Films containing green-tea polyphenols showed a significant antioxidative capacity, and they can be used for protection of food containing a high percent of fats.	Wrona et al. [136]
Tea polyphenols	Gelatin	Not reported	CS NPs provided controlled-release of tea polyphenols, and this increased its antioxidant properties. EGCG-loaded nanocomplex can be used for protection of fatty foods.	Liu et al. [143]
Epigallocatechin gallate (EGCG)	Zein/chitosan NPs	155.5–240.6 nm	EGCG-loaded zein/chitosan NPs possess high antioxidant activity and can be applied against degradation and oxidation of fatty foods; moreover, in the future, these nanocomplexes can be applied as active material for edible films in the food industry.	Liang et al. [135]
Gallic acid	Zein ultra-fine fibers	327–387 nm	Gallic acid retained its antioxidant activity after incorporation into zein ultra-fine fibers, and thus this prepared ingredient can find application in packaging materials.	Neo et al. [8]
Rosemary (*Rosmarinus officinalis*) polyphenols	Polyvinyl alcohol (PVA) electrospun nanofibers	307 ± 33 nm/282 ± 39 nm	PVA active mats successfully incorporated bioactive components from rosemary extract, showing an excellent antioxidant activity. This may find application for active food packaging, especially for hydrophilic and acid food products.	Estevez-Areco et al. [144]

The physicochemical properties of these particles can create a pro-oxidant environment in cells, which leads to the creation of free radicals and inflammation or even cell death [145]. Different in vivo and in vitro studies showed that exposure to nanoparticles may affect epigenetic processes, such as DNA methylation, histone modifications, and RNA interference [146], and it is certain that nanoparticles can cause DNA damage [147].

It has been suggested that the cytotoxicity of gold nanoparticles depends mostly on their size [148]. Another study on silver nanoparticles emphasized that size is a very important property because of the cellular uptake of these particles through pores [149]. A few of the most important silver nanoparticles and genotoxicity responses were listed by Manickam et al. [150]. The majority of studies showing nanoparticle toxicity are linked to their use in medicine and pharmacology, while risks concerning food consumption have not been sufficiently studied.

Materials that are harmless to humans may show toxicity if they are nanosized, which leads to doubts whether nanomaterials deserve special regulative frames. A summary of EU and some non-EU countries’ legislations about the use of nanomaterials can be found [151]. Nanomaterial toxicity, behavior, and bioaccumulation are the main uncertainties because of the lack of sufficient scientific data. This must be overcome, and studies must be carried out in risk-and-exposure assessments regarding the use of nanomaterials [152]. The European Food Safety Authority (EFSA) has published guidance on risk assessment on this topic [153]. This guidance is a result of a detailed revision of the previous version, regarding certain nanospecific aspects. It suggests that the existing definition of engineered nanomaterial should not define the boundaries of risk assessment, which also addresses other type of materials, and it emphasizes the importance for future research to fill the gaps in order to perform accurate nanomaterial safety assessment.

The European Regulation on Registration, Evaluation, Authorization, and Restriction of Chemicals (REACH) has a special committee dealing with nanomaterials [154]. In the United States, the National Organic Standards Board recommended that engineered nanomaterials should be prohibited from food products bearing the USDA Organic label [155].

### 4.2. Polyphenol Toxicity

The biological activity of polyphenols is mostly related to their ability to chelate metals and scavenge free radicals, but they can exhibit pro-oxidant behavior under certain conditions, leading to the formation of reactive oxygen species that can damage DNA, lipids, and other biomolecules [156]. Antioxidant/prooxidant activity depends on several factors, such as the presence of redox active chemicals, biological-tissue pH, and solubility characteristics [156]. It has been shown that consumption of tea in high doses leads to an imbalance in the antioxidant and pro-oxidant behavior of tea flavonoids, thus resulting in detrimental effects to human health, including the hepatotoxicity of catechins from green tea, the reduction of intestinal absorption of dietary iron, and the precipitation of digestive enzymes by tannins from black tea, and the reduction of lipase activity by polyphenols from oolong tea [138]. According to a scientific opinion on the safety of green-tea catechins given by the EFSA ANS Panel [157], the consumption of green-tea catechins from green-tea infusions and similar drinks is generally safe, but the intake of EGCG as a food supplement in doses equal to or above 800 mg/day for four months and longer statistically significantly increases serum transaminases in human blood, which indicates liver injury.

Besides pro-oxidant activity, the complexation of bioelements by polyphenol compounds was also observed, which can negatively impact human health. It has been reported that phenolic acids, cyanidin derivatives, delphinidin, quercetin, kaempferol, morin, epigallocatechin-3-gallate, and curcumin are capable of binding bioelements (Fe, Mg, Mn, Zn, Se, Co, and Cu), and thus decrease their absorption in the gastrointestinal tract and, hence, their content in blood and tissue, which can lead to a disorder of bioelement-dependant metabolic pathways [158].

The general conclusion is that nanomaterials are not a uniform group when it comes to risks, even though they have many potential beneficial applications in the food sector; before utilization, they have to pass serious tests. Furthermore, the intake of polyphenols in high doses for a long period of time has harmful effects to humans. The toxicity of polyphenols encapsulated as nanoparticles is still not reported, and there is evidence that the coadministration of polyphenols (quercetin) with ENPs (silver nanoparticles) reduces the harmful effects of ENPs, such as cytotoxicity and oxidative stress [159]. However, further studies should be conducted to estimate the potential toxic effects of polyphenol-loaded nanoparticles.

## 5. Conclusions

The development of nanosized functional food ingredients containing polyphenols has resulted in improvements in food safety and quality, smart packaging, the targeted delivery of compounds, and the improved sensory properties of food product. Most frequently encapsulated polyphenols are catechins, quercetin, eugenol, epigallocatechin, EGCG, curcumin, and tea polyphenols. Nanoparticles such as cyclodextrins, gelatin, casein and whey proteins, zein, chitosan, or complex nanoparticles are generally used as nanocarriers. These carriers protect sensitive polyphenols and preserve beneficial polyphenolic properties. Thanks to their nanosize, the delivery of these particles is upgraded. However, the possible toxicity of nanomaterials, such as ENPs, which are used in food-packaging materials, should be addressed, and future work should be focused on the validation of relevant methods for the characterization of nanomaterials in complex matrices, and for measuring their reactivity and in vitro degradation, in order to facilitate safety-risk assessments. Another approach could be the use of native nanocarriers, such as casein micelles, for which the GRAS status is verified. It should be kept in mind that numerous health claims related to polyphenols have failed to be demonstrated in vivo studies, and the intake of polyphenols may pose health risks in high doses. Certainly, knowledge of nanotechnology and polyphenols and awareness of their benefits and risks should be increased.

## Figures and Tables

**Table 1 nanomaterials-09-01629-t001:** Nanoencapsulation of phenolics important for food processing.

Active Compounds	Nanocarriers	Particle Size (nm)	Activity (Details of Research)	Reference
**Polyphenol-Loaded Nanoparticles for Enhancement of Physicochemical Properties of Food**				
Curcumin	Zein-nanoparticles (NPs)	175–900 nm	The nanoparticles showed good dispersion and coloring capacity in semi-skimmed milk compared to commercial curcumin. The nanoparticles thus enable the use of curcumin as a coloring agent in aqueous food products.	Gomez-Estaca et al. [31]
Curcumin	Nanomicelles	30 nm	Nanomicelles (natural colorants) allowed better intestinal resorption of active compounds and enhanced their stability.	Ranjan et al. [5] Silva et al. [51]
Herb essential oils (containing high percent of phenolic terpenes)	Nanoemulsion based on herb essential oils	59.48–112.82 nm	Essential oil nanoemulsions enhanced organoleptic quality of rainbow trout and effectively affected the reduction of bacterial growth.	Ozogul et al. [14]
**Polyphenol-Loaded Nanoparticles for Enhancement of Functional Properties of Food**				
Curcumin	Pectin-coated sodium caseinate/zein NPs	250–600 nm	Curcumin-loaded nanoparticles significantly enhanced curcumin antioxidant activity and prolonged release capabilities in simulated gastric and intestinal fluids.	Chang et al. [43]
Curcumin	Caseinate-zein-polysaccharide nanocomplex	160–210 nm	Nanocarriers exhibited good physicochemical properties and possibility for future applications as oral delivery vehicles for lipophilic nutrients.	Chang et al. [52]
Curcumin	Chitosan-coated solid-lipid NPs	451.8 ± 19.62 nm	Chitosan-coated solid-lipid nanoparticles as carriers for curcumin contributed to increased oral bioavailability and affected the wider application of curcumin nanostructures in food.	Ramalingam et al. [53]
Catechin	chitosan/poly-γ-glutamic acid NPs	Not reported	Chitosan/poly-γ-glutamic acid nanoparticles enhanced the oral delivery of catechins and improved antioxidant activity of catechins.	Tang et al. [54]
Catechin	Gelatin NPs	Around 200 nm	Catechin–gelatin nanoparticles can be a useful antioxidant carrier because catechin and gelatin are mutually protected from oxidation and enzymatic degradation.	Chen et al. [36]
(+)-catechin (−)-epigallocatechin	Chitosan nanoparticles (CS NPs)	˂500 nm	Encapsulation of catechins in CS NPs enhanced catechins’ intestinal absorption and their bioavailability.	Dube et al. [55]
Catechin	Bioadhesive CS NPs	110–130 nm	Encapsulation of catechins in CS NPs leads to enhanced oral bioavailability of catechin.	Dudhani & Kosaraju [56]
Tea polyphenols (TP)	CS NPs (using carboxymethyl chitosan and chitosan hydrochloride)	407 ± 50 nm	CS-TP NPs showed significant antitumor activities.	Liang et al. [57]
Quercetin	Chitosan/alginate NPs	Not reported (˃100 nm)	Chitosan/alginate nanoparticles can be good carriers for quercetin because of their safe and improved protection of the encapsulated antioxidant.	Aluani et al. [58]
Quercetin	Poly-D,L-lactide (PLA) NPs	130 ± 30 nm	Encapsulation of quercetin in poly-D,L-lactide (PLA) nanoparticles affected the increased solubility and stability of the quercetin.	Kumari et al. [59] Esfanjani & Jafari [2]
Quercetin	Solid-lipid nanoparticles (SLNs)	155.3 nm	SLNs can be considered appropriate oral delivery carriers for poorly water-soluble quercetin because they enhanced their absorption.	Li et al. [60]
(−)-epigallocatechin gallate (EGCG)	Chitosan-tripolyphosphate nanoparticles (CS NPs)	440 ± 37 nm	CS NPs can be useful carriers, providing better oral delivery and stability of EGCG.	Dube et al. [61]
(−)-epigallocatechin gallate (EGCG)	Chitosan/β-lactoblobulin (β-Lg) NPs	100–500 nm	The prolonged release capabilities of EGCG-loaded chitosan/β-Lg nanoparticles affected the increase of effective absorption of EGCG in the human intestine.	Liang et al. [45]
(−)-epigallocatechin gallate (EGCG)	Chitosan-caseinophospho-peptide nanocomplexes (CS-CPP)	150 ± 4.3 nm	CS-CPP nanocarriers influenced the enhancement of intestinal permeability of EGCG.	Hu et al. [50]
Rutin	Casein/pectin nanocomplex	Not reported	Sodium caseinate-pectin nanoparticles have high potential for oral delivery nutrients. They showed limited release of rutin in simulated intestinal conditions.	Luo et al. [49]
Naringenin	β-casein NPs	˂100 nm	The research results suggested that naringenin binds with β-casein over Van der Waals forces, hydrogen bonds, and hydrophobic interactions, which improved individual functional characteristics of naringenins, primarily by enhancing their solubility.	Moeiniafshari et al. [62]
Phenolics of pomegranate peel	Lyophilized pomegranate peel-nanoparticles	Not reported	Lyophilized pomegranate peel-nanoparticles demonstrated effective antioxidant and antimicrobial properties, improving cooking characteristics of meatballs and prolonged quality of meatballs during storage.	Morsy et al. [63]
Guabiroba fruit phenolic extracts	Poly(D,L-lactic-co-glycolic)acid NPs(PLGA)	202.5–243.8 nm	PLGA can be used as a nanocarrier for phenolic compounds. Loaded-nanoparticles have showed inhibitory effect on *Listeria innocua* and good antioxidant activity.	Pereira et al. [64]
Rosmaric acid, protocatechuic acid, 2,5-dihydroxybenzoic acid	CS NPs	˃300 nm	Polyphenol-loaded chitosan nanoparticles showed effect against food pathogens. Better antimicrobial activity was obtained against *Escherichia coli O157:H7* and *Bacillus cereus*, while the effect against *Salmonella typhimurium* was less pronounced.	Madureira et al. [32]
Thymol	Zein NPs-stabilized with sodium caseinate and chitosan hydrochloride	204.75 nm	Thymol-loaded nanoparticles had strong activity against *S. aureus* and other Gram-positive bacterium under the experimental conditions and can be used as nanocarriers for antimicrobial agents in food.	Zhang et al. [65]
Thymol/carvacrol	Thymol/carvacrol liposomes (TCL)	270.2 nm	TLC can be used to suppress biofilm formation in the early stages of bacterial attachment to food-contact surfaces and it showed antimicrobial activity against *S. aureus* and *Salmonella*. Application of TLC presents a good perspective for food quality and safety improvement.	Engel et al. [66]
Thymol/carvacrol	Zein NPs	108–122 nm	Phenolic monoterpenes give a strong interaction with wall of zein. This phenolic-loaded NPs showed higher antimicrobial activity and phenolics remained stable during storage and food processing.	Da Rosa et al. [67]
Eugenol	Zein/sodium caseinate/pectin complex NPs	140 nm	Eugenol-loaded colloidal nanoparticles can find application in the food industry as a dry powder formulation with antimicrobial properties.	Veneranda et al. [48]
Eugenol	Sesame oil blended eugenol-loaded nanoemulsion	13–191 nm	Nanoemulsion exhibited activity against S. aureus and affected the reduction of heterotrophic bacteria in orange juice, and it can be used for food preservation (against microbial spoilage).	Ghosh et al. [68]
Eugenol	Nanoemulsions (using gum arabic and lecithin)	103.6 ± 7.5 nm	Eugenol-loaded nanoemulsion possesses powerful antimicrobial properties and can be applied in the food industry as a food preservative.	Hu et al. [69]
EGCG	Nanostructured lipid carriers (NLC) functionalized with folic acid	234–359 nm	The developed formulation of nanoencapsulated EGCG was suitable for the oral delivery and has potential for applications in the food industry.	Granja et al. [70]
EGCG and EGCG + piperine	Zein	118.3 and 184.2 nm	Optimization of nanoformulation of EGCG alone and along with piperine into a protein nanocarrier and the study of their effect on in vitro antioxidant, hemolytic, and anticancer activities.	Dahiya et al. [71]
The fruit extract of Ribes nigrum	Silver nanoparticles (Ag-NPs)	5–10 nm	Efficiency of nanoencapsulation, characterization and bactericidal, fungicidal, and anticancer activities of nanoparticles synthesized using the fruit extract of Ribes nigrum.	Dobrucka et al. [72]
Curcumin and quercetin	Re-assembled casein micelles (r-CM) and casein nanoparticles (CNPs)	186.9, 66.2, 72.8, and 186.5 nm	Both CNP and r-CM significantly improved the chemical stability of phenolic compounds, and the aqueous solubility was higher than that of free molecules.	Ghayour et al. [73]
Resveratrol	Zein and zein + alginate/chitosan complex coating	72 nm and 160.9 nm	Alginate/chitosan-complex coating improved the photostability, sustained release and bioaccessibility of resveratrol and could be suitable delivery system.	Khan et al. [74]
Orange oil nanoemulsions	Orange oil, carrier oil, nonionic surfactant	25–100 nm	Orange oil as a lipophilic functional agent was successfully incorporated into nanoemulsions; the influence of surfactant, oil composition, temperature, and storage stability were evaluated.	Chang & McClements [75]
Thyme oil nanoemulsions	Thyme oil-in-water nanoemulsions	120 and 1300 nm	Thyme oil was used as a core for preparation of antimicrobial system tested against acid-resistant spoilage yeast, Zygosaccharomyces bailii.	Chang et al. [76]

**Table 2 nanomaterials-09-01629-t002:** Methods used for determination of the biological properties of polyphenol-loaded nanoparticles.

Antioxidant/Cytotoxic (Cell line/Animal Model) Assays	Active Compounds	Nanocarrier	Reference
ABTS radical scavenging activity	Curcumin	Pectin-coated sodium caseinate/zein NPs	Chang et al. [52]
	Tea polyphenols	Lysozyme-carboxymethyl cellulose nanogels	Liu et al. [103]
	Catechin	Chitosan/poly-γ-glutamic acid NPs	Tang et al. [54]
	Curcumin	Caseinate-zein-polysaccharide nanocomplex	Chang et al. [43]
DPPH radical scavenging activity	Resveratrol	Chitosan-TPP (sodium tripolyphosphate) NPs	Wu et al. [104]
	Resveratrol/quercetin	Chitosan NPs/polyethylene glycol modified chitosan NPs	Natesan et al. [105]
	Resveratrol	PLGA [poly(lactic-co-glycolic acid)] -oil hybrid NPs	Kumar et al. [106]
	Resveratrol/quercetin	Liposome	Caddeo et al. [107]
	Tea polyphenols	Lysozyme-carboxymethyl cellulose nanogels	Liu et al. [103]
	Catechin	Chitosan/poly-γ-glutamic acid NPs	Tang et al. [54]
Hypochlorous acid (HOCl) scavenging assay	Resveratrol	Bovine serum albumin-based NPs	Fonseca et al. [108]
Ferric-reducing ability (FRP)	Resveratrol/quercetin	Chitosan NPs/polyethylene glycol modified chitosan NPs	Natesan et al. [105]
Hydrogen peroxide scavenging assay	Resveratrol	PLGA [poly(lactic-co-glycolic acid)] -oil hybrid NPs	Kumar et al. [106]
TEER measurements and transport studies (Caco-2 cell)	Catechin	Chitosan/poly-γ-glutamic acid NPs	Tang et al. [54]
Cell viability (Hepatocellular carcinoma cells SMMC7721 and hepatocyte LO2 cells)	Resveratrol	Chitosan-TPP (sodium tripolyphosphate) NPs	Wu et al. [104]
Monkey kidney (Vero) cell lines-sulforhodamine B assay	Resveratrol	PLGA [poly(lactic-co-glycolic acid)] -oil hybrid NPs	Kumar et al. [106]
Antitumor effect in vitro assays (CT26 mouse colon cancer cells)	Resveratrol	Polyethylene glycol-polylactic acid polymer NPs	Jung et al. [109]
In vitro cell culture study (Cochlear cell lines (HEI-OC1 and SVK-1)	Resveratrol	Polymeric NPs	Musazzi et al. [110]
In vitro hemolytic/anticancer assay (human cancer cell lines i.e., leukemia cancer (HL60), oral cancer (SCC40), breast cancer (MCF7), cervix cancer (HeLa) and colon cancer (Colo205)- sulforhodamine B assay)	EGCG/EGCG + piperine	Zein	Dahiya et al. [71]
In vitro cytotoxicity assay (SK-MEL-28 and Colo-38 cells)	Resveratrol	Ultradeformable liposomes	Cosco et al. [111]
In vitro assays in cells from different origin (cultivated HepG2 cells, isolated primary rat hepatocytes, isolated murine spleen lymphocytes and macrophages)	Quercetin	Chitosan/alginate NPs	Aluani et al. [58]
In vitro cytotoxicity assay (Human hepatoma HepG2 cells)	Tea polyphenols (TP)	CS NPs (using carboxymethyl chitosan and chitosan hydrochloride)	Liang et al. [57]
In vitro assay (human hepatoblastoma cancer cell line HepG2)	Tea polyphenols	Lysozyme-carboxymethyl cellulose nanogels	Liu et al. [103]
Evaluation of cell proliferative activity (nonmalignant line of fibroblasts CCD-39Lu-isolated from lungs and adherent epithelial non-small cell lung cancer cell line A549)	The fruit extract of Ribes nigrum	Silver nanoparticles (Ag-NPs)	Dobrucka et al. [72]
In vitro assay (Human dermal fibroblasts)	Resveratrol/quercetin	Liposome	Caddeo et al. [107]
Antitumor effect in vivo assays (CT26 mouse colon cancer cells)	Resveratrol	Polyethylene glycol-polylactic acid polymer NPs	Jung et al. [109]
IOP reducing efficiency (normotensive rabbits)	Resveratrol/quercetin	Chitosan NPs/polyethylene glycol modified chitosan NPs	Natesan et al. [105]
In vivo study of antidiabetic activities (Wistar rats)	Curcumin	Chitosan CS-ZnO-NC NPs	Chauhan et al. [112]
In vivo toxicological evaluation (Male Wistar albino rats)	Quercetin	Chitosan/alginate NPs	Aluani et al. [58]

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
