# Peer review of "Application of Polyphenol-Loaded Nanoparticles in Food Industry"

_nanomaterials, 2019, doi:10.3390/nano9111629_

Round 1
Reviewer 1 Report
The manuscript reports an overview of the use of nanoparticles, specifically polyphenol-loaded nanoparticles in food industry. Different applications are reported. Aspects related to the health-promoting properties as well as toxicity are also included. In addition, applications in food packaging are comment on as well.
The paper is well- organized and the objectives well defined.
However, some aspects could be improved, particularly, in section 3 related to packaging, some applications of smart packaging should be included. In section 4 Toxicity of nano-based materials, it would be convenient include some aspects such as the migration of nanoparticles to food and comment on.
I do express my positive opinion on the acceptance of the manuscript to be published after minor revision.
Author Response
REVIEWER 1
The manuscript reports an overview of the use of nanoparticles, specifically polyphenol-loaded nanoparticles in food industry. Different applications are reported. Aspects related to the health-promoting properties as well as toxicity are also included. In addition, applications in food packaging are comment on as well.
The paper is well- organized and the objectives well defined.
However, some aspects could be improved, particularly, in section 3 related to packaging, some applications of smart packaging should be included. In section 4 Toxicity of nano-based materials, it would be convenient include some aspects such as the migration of nanoparticles to food and comment on.
I do express my positive opinion on the acceptance of the manuscript to be published after minor revision.
Many thanks for your comments and suggestions. The additional text is included into the manuscript. In the section 3, lines 257-261 and in the section 4, lines 289-297 marked in red letters.
Reviewer 2 Report
Perfect review of this topic
BR
Author Response
REVIEWER 2
Perfect review of this topic.
Many thanks for your opinion.
Reviewer 3 Report
The manuscript is very interesting and well organised. I would suggest to the authors to add the description of the main methods used for characterize the biological properties of the loaded nanoparticles.
In the introduction please review the following articles:
Giosafatto et al. Effect of Mesoporous Silica Nanoparticles on Glycerol-Plasticized Anionic and Cationic Polysaccharide Edible Films. Coatings 2019, 9, 172; doi:10.3390/coatings9030172 .
Mangiacapra, P.; Gorras, G.; Sorrentino, A.; Vittoria, V. Biodegradable nanocomposites obtained by ball milling of pectin and montmorillonites. Carbohydr. Polym. 2006, 64, 516–523.
Author Response
REVIEWER 3
The manuscript is very interesting and well organised. I would suggest to the authors to add the description of the main methods used for characterize the biological properties of the loaded nanoparticles.
Thank you for your comments and suggestion. The description of main methods is added into the manuscript as table 2, line 182- and as text, lines 178-181; 251-253.
In the introduction please review the following articles:
Giosafatto et al. Effect of Mesoporous Silica Nanoparticles on Glycerol-Plasticized Anionic and Cationic Polysaccharide Edible Films. Coatings 2019, 9, 172; doi:10.3390/coatings9030172 .
Mangiacapra, P.; Gorras, G.; Sorrentino, A.; Vittoria, V. Biodegradable nanocomposites obtained by ball milling of pectin and montmorillonites. Carbohydr. Polym. 2006, 64, 516–523.
These references were added into the Introduction, line 54 as references 15 and 16.